# Benzoxazine Containing Fluorinated Aromatic Ether Nitrile Linkage: Preparation, Curing Kinetics and Dielectric Properties

**DOI:** 10.3390/polym11061036

**Published:** 2019-06-11

**Authors:** Sijing Chen, Dengxun Ren, Bo Li, Kui Li, Lin Chen, Mingzhen Xu, Xiaobo Liu

**Affiliations:** Research Branch of Advanced Functional Materials, School of Materials and Energy, University of Electronic Science and Technology of China, Chengdu 610054, China; csj20189@126.com (S.C.); rendenxun2008@126.com (D.R.); 14704177593@163.com (B.L.); leekingue2014@163.com (K.L.); linchen_uestc@163.com (L.C.)

**Keywords:** benzoxazine, aromatic ether nitrile linkage, curing kinetics, thermal stability, dielectric properties

## Abstract

Benzoxazine containing fluorinated aromatic ether nitrile linkage (FAEN-Bz) had been synthesized from 2,6-dichlorobenzonitrile, 4,4’-(hexafluoroisopropylidene)diphenol (bisphenol AF), 3-Aminophenol, formaldehyde, phenol by condensation polymerization and Mannich ring-forming reaction. Structures of the monomer were verified by Proton NMR spectrum (^1^H-NMR) and Fourier transform infrared spectroscopy (FTIR). Curing behaviors and curing kinetics of designed monomers were investigated and discussed. The activation energy was calculated and possible polymerization mechanisms were also proposed. Then, properties of cured polymers including crosslinking degrees, thermal decomposition, surface wettability and energy, and dielectric properties were studied and discussed. Additionally, programmed integral decomposition temperature (IPDT) was also used to evaluate the thermal stability of final polymers. Results indicated that the incorporation of benzoxazine and nitrile resulted in increased thermal stability and char yields. Moreover, the surface wettability and dielectric properties of poly(FAEN-Bz) can be easily controlled by tuning the curing temperatures and time.

## 1. Introduction

With the development of science and technology, thermosetting resins play an important role in electronics, substrate materials, aeronautics-aerospace industry and new energy applications. Thermosetting resins, like epoxy, cyanate ester, phthalonitrile, bismaleimide and so on, due to the advantages including high specific strength and compressive strength, corrosion resistance, electrical properties, light weight, high glass transition temperature (*T*_g_) and outstanding bonding properties were intensively investigated and applied in many high-tech areas [1,2,3,4,5]. For applications in particular areas and human encountered with energy crisis, polymer-based materials with easy processing were widely used to replace the conventional metal materials to reduce the weight and decrease the costs [6,7]. However, common thermal stability (150–300 °C) and low glass transition temperature (80–180 °C) of traditional polymers limit their further applications in the fields of aerospace, ocean, car industry, new energy and electronic component packaging.

Phenolic resin possesses many good properties, such as low cost, dimensional stability, heat resistance, electrical insulation, flame retardant and so on. It has been widely used in construction, electronics, aerospace and other fields [8]. However, traditional phenolic resin often releases volatiles which corrode the processing equipment during the curing process [9]. It will lead to the formation of micro-voids, and damage the performance of final materials. Benzoxazine resin is generated from traditional phenolic resin which not only retains advantages of common phenolic resins, such as high thermal stability, good mechanical properties, and better electrical properties, but also has some unique properties superior to phenolic resins, such as rich molecular design flexibility, no small molecule release during the crosslinking process, close to zero volume contraction, low water absorption, and high *T*_g_ and high char yield [10]. 

In our previous work, novel kinds of benzoxazine with various active groups were designed and fabricated [11,12]. All of the previous work indicated that the benzoxazine-based resin containing nitrile groups and allyl possessed satisfactory properties in curing processing and structural applications. For example, allyl-functional phthalonitriles-containing benzoxazine and phthalonitrile-containing benzoxazine were successfully prepared with excellent properties like wide process temperature (~80 °C), high modulus, high *T*_g_ (> 350 °C) and high thermal stability (*T*_5%_ > 450 °C in N_2_). Unfortunately, benzoxazine-base resin is usual brittleness, insufficient toughness, poor resistance to impact and stress cracking, which limit its wide applications in structural materials with high toughness [13,14]. To improve the brittleness of thermosetting composites, various methods were attempted, including the physical blending of thermoplastic and thermosetting resins, the chemical bonding of long molecular segments into the thermosetting resin by molecular designing. High-performance thermoplastic polymers such as poly(ether imide), polyaryletherketone, poly(arylene ether nitrile) and amines (octanediamine and *m*-xylylenediamine) were blended with benzoxazine resins to enhance the toughness [15,16,17,18]. Results indicated that the brittleness of the composites was improved, but the incompatible also led to the decrease in thermodynamic performance. Another efficient way to improve the toughness was to synthesize the benzoxazine resin with intrinsic flexible molecular linkages. In this way, the prepared benzoxazine resin not only possessed the improved toughness, but also can maintain the thermodynamic performance. Chen and their co-workers reported three main-chain type benzoxazine polymers with improved toughness were synthesized via bishydroxydeoxybenzoin and three kinds of aromatic diamines, respectively. As the results showed that the thermal stability were maintained [19]. Thus, increasing researches focused on the molecular designing to improve the final properties of thermosetting resins.

Recently, materials with high performance and low dielectric constant have attracted wide attentions of researchers. Thermosetting and thermoplastic materials containing fluorine were designed and prepared. The researchers investigated various kinds of benzoxazine resins containing fluorinated groups such as bisphenol-AF [20], 1,4-tetrafluorobenzene or 4,4-octafluorobiphenylene dioxyphenylene [21,22] in molecular chains. The results indicated that the introduction of fluorine can serve the reducing of dielectric constants and dielectric loss. It was proposed that electron cloud density of fluorine is high, so the polarizability of fluorine is low when polarized by external electric field. At the same time, fluorine-containing groups with larger volume were introduced into polymer molecules to increase the free volume between molecular chains to reduce the dielectric constants and loss. Moreover, the introduction of fluorinated groups can improve the thermal properties of polymers to a certain extent because of the high binding energy of C–F bonds (489 kJ/mol) [23,24].

Polymer bisphenol A-based poly(arylene ether nitrile) was a kind of high performance thermoplastics, due to the aromatic ether nitrile linkage the polymer exhibited a high *T*_g_ (>170 °C), outstanding tensile strength (>90 MPa) and excellent thermal stability (*T*_5%_ > 480 °C in N_2_) [17,25]. Herein, in this work, a molecular designing method was used to prepare modified benzoxazine containing fluorinated aromatic ether nitrile linkage. The molecular structures, its curing behaviors, kinetics, and possible polymerization mechanisms were investigated and discussed. Also, properties including thermal stability, surface wettability and dielectric properties were analyzed to discuss the influence of the fluorinated aromatic ether nitrile linkage on the properties of benzoxazine-based polymers.

## 2. Experimental

### 2.1. Material

4,4′-(Hexafluoroisopropylidene)diphenol (bisphenol AF) and 2, 6-dichlorobenzonitrile were purchased from Tianjin BODI Chemicals, Tianjin, China. 3-Aminophenol (purity 98%) was provided by Aladdin Industrial Corporation, Shanghai, China. N, N-dimethylacetamide (DMAc, AR), ethanol (AR), toluene (AR), potassium carbonate anhydrous (K_2_CO_3_, AR), paraformaldehyde (AR), phenol (AR), acetone and methanol were obtained from Chengdu Kelong chemicals Corporation, Ltd., Chengdu, China. All of the chemicals were used as received without further purification.

### 2.2. Synthesis of the Amino-Terminated Fluorinated Aromatic Ether Nitrile Linkage (FAEN-NH_2_) Monomers

The amino-terminated fluorinated aromatic ether nitrile linkage was synthesized according to our previous work with minor modifications [26,27]. The synthetic route was presented in Scheme 1. In Scheme 1a, chlorinated terminated monomers (FAEN-Cl) were first synthesized in solvent of DMA_C_ and toluene (volume ratio 3:1), from 2,6-dichlorobenzonitrile and bisphenol AF in the presence of K_2_CO_3_. This synthetic procedure was as follows. The solvent DMA_C_ and toluene, 2,6-dichlorobenzonitrile, bisphenol AF and K_2_CO_3_ were added in three flasks with a mechanical stirring and refluxing condenser. The reaction was kept at 155–160 °C for 2 h to remove the water and toluene. After that, the temperature gradually increased to 170 °C for 1 h. Then the system was cooled to 100 °C, 3-Aminophenol was added into the solution and keept at 90 °C to prepare amino terminated monomers (FAEN-NH_2_). The structures and prparation process were shown in Scheme 1b. The obtained mixed solution was precipitated in deionized water. Then, the precipitation was purified several times with deionized water and dried in a vacuum oven at 60 °C for 10 h.

### 2.3. Preparation of Bbenzoxazine Containing Fluorinated Aromatic Ether Nitrile (FAEN-Bz) Monomers

The FAEN-NH_2_ monomers were added into a bottle with moderate phenol and paraformaldehyde, and ethanol and toluene was added as the solvent. Then, raising the temperature slowly to 80–85 °C and maintained for 5 h to get FAEN-Bz solution. After that, the prepared FAEN-Bz with solution was placed into oven at 110 °C for 4 h to remove the solvents to obtain the FAEN-Bz resin [12]. The detailed synthetic route was shown in Scheme 1c.

### 2.4. Preparation of Benzoxazine Containing Fluorinated Aromatic Ether Nitrile Polymers (poly(FAEN-Bz))

FAEN-Bz monomers were dissolved in DMAc (*V*_DMAc_:*m*_FAEN-BZ_ = 1:1) with stirring at 50 °C for 30 min. Then, the FAEN-Bz solution was cast onto a clean glass plate in an oven with the sequential temperature procedure at 80 °C for 1 h, 100 °C for 1 h, 120 °C for 1 h, 140 °C for 1 h, 160 °C for 2 h and 180 °C for 2 h to remove the DMAc solvent. The basic curing program was carried out 200 °C for 4 h. To study the effects of curing degree on final properties, FAEN-Bz was heat treated at various temperature conditions, including 220 °C for 2 h, 240 °C for 2 h, 260 °C for 2 h, and 280 °C for 2 h.

### 2.5. Characterizations

Fourier transform infrared spectroscopy (FTIR) spectra was recorded with Shimadzu FTIR 8400S (Shimadzu, Kyoto, Japan) in KBr pellets. In situ FTIR spectra was characterized by PerkinElmer Spectrum 200 (PerkinElmer, Waltham, MA, USA) with a heating rate of 10 °C/min from 50 °C to 300 °C. Proton NMR (1H-NMR) spectra were obtained by Bruker AV400 nuclear magnetic resonance (NMR, Bruker, Karlsruhe, Germany) spectrometer at a proton frequency of 400 MHz. Curing behaviors of FAEN-Bz were studied by differential scanning calorimetric (DSC, Q100, TA Instruments, Newcastle, DE, USA) under a nitrogen atmosphere with a flow ratio of 50 mL/min. Gelation time was performed by Shanghai Yijia Electrical Co. Gel time tester (YJ139-LA38-11BN), Shanghai, China. Thermal gravimetric analysis (TGA) tested on TA Instruments Q50 (TA Instruments, Newcastle, DE, USA) with a heating rate of 20 °C/min under nitrogen from 50 °C to 800 °C. The fracture surface morphology of the polymer was scanned though scanning electron microscope (SEM, JSM25900LV, JEOL, Akishima, Japan) operating at 20 kV. Small angle X-ray scattering (SAXS, Bruker AXS D8, Karlsruhe, Germany) was applied to investigate the phase morphology of cured FAEN-Bz. Surface wettability was characterized with JY-PHa Contact Angle Tester (Chengde youte Testing Instrument Co., Ltd., Chengde, China). Dielectric properties were tested by TH 2826 LCR meter (Tonghui Electronic Co., Ltd., Changzhou, China), which was carried out at different frequencies (500 Hz–5 MHz) at room temperature.

## 3. Results and Discussion

### 3.1. Characterization of the Structures of FAEN-NH_2_ and FAEN-Bz Monomers

The molecular structure of FAEN-NH_2_ monomer and FAEN-Bz were investigated by ^1^H-NMR (Figure 1) and FTIR (Figure 2) spectroscopy. In Figure 1a, the resonance peak appearing at 3.76 ppm was assigned to the protons of amino groups. Resonance appearing at 6.82 and 7.46 ppm were protons from aromatic rings with nitrile groups. In Figure 1b, resonances appearing at 4.64 and 5.35 ppm were assigned to the methylene protons in oxazine rings [28]. Similarly, the aromatic protons with nitrile groups were observed at 6.82 and 7.46 ppm, indicating that the designed structures were obtained.

Figure 2 shows the FTIR spectra of FAEN-NH_2_ and FAEN-Bz. In Figure 2a, it was obvious that the intensive absorption peak of nitrile groups appeared at about 2230 cm^−1^ [29]. An absorption band of –C–F and –C–O–C of asymmetric stretching oscillations appeared at 1200 and 1245 cm^−1^, respectively. Characteristic absorption band of –NH_2_ was observed at 3460 and 3381 cm^−1^ [30]. In Figure 2b, the characteristic absorption band of antisymmetric oxazine ring stretch appeared at 953 cm^−1^ [16,31]. Moreover, a wide absorption band at around 3412 cm^−1^ was observed, which can be assigned to the intermolecular association of hydroxyl bonds, generated from the trace hydrogen and nitrogen atoms in benzoxazine rings. To sum up, results of ^1^H-NMR and FTIR spectroscopy confirmed the designing structures of FAEN-NH_2_ and FAEN-Bz.

### 3.2. Curing Kinetics of FAEN-Bz Monomers

#### 3.2.1. Gelling Analysis of FEAEN-Bz Monomers

For thermosetting resin matrix, gel time was usually used to reflect the reaction activity and curing behaviors. In this work, the gel time of FAEN-Bz at various temperatures were studied and results were presented in Table 1. It can be seen that with increasing test temperatures, the gel time significantly decreased. Moreover, when the temperature was 190 °C, gel time of FAEN-Bz was less than 3 min, which had direct meaning to the preparation of FAEN-Bz polymer.

According to the Arrhenius equation, the relationship between gel time (t_gel_) and reaction activity (E_α_) was presented as Equation (1).
lgt_gel_ = E_α_/(2.303RT) + A(1)

From the equation, a plot of lgt_gel_ vs. 1/T_α_ would result in a straight line with a slope of E_α_/2.303R. The plot of lgt_gel_ as function of 1/T_α_ was shown in Figure 3. A good linear relationship is observed from the plot and the average energy values (E_α_) of FAEN-Bz was calculated as 98.7 kJ/mol from the slope of the straight. The average activation energy value was slightly lower than that of benzoxazine (102–116 kJ/mol) reported previously [32,33]. It was well known that the polymerization of oxazine rings began along with that the oxygen or nitrogen atoms in oxazine rings were attacked by active hydrogen. Thus, increasing the electron cloud around oxygen or nitrogen atoms was in favor to increase the activity of benzoxazine-based monomers [34]. For FAEN-Bz, it can be attributed to the fact that electron withdrawing groups (nitrile groups and fluorine atoms) have induced the *π* electron cloud of benzene slightly shifted to O and N atoms in oxazine rings, which accelerated the polymerization of FAEN-Bz.

#### 3.2.2. Kinetic Analysis by Differential Scanning Calorimetric (DSC)

Curing behavior tested with DSC could provide the data necessary for calculating the time/temperature-dependent conversion of kinetic parameters. Figure 4 showed DSC curves of FAEN-Bz at various heating rates. There was an exothermic peak in all of the curves, which indicated that FAEN-Bz possessed one polymerization process. With increasing the heating rate, exothermic peak became higher and moved to higher temperature range, which was dominated by the instrument sensitivity and was accredited.

According to our previous work [32,35], conversion rates (α) were calculated and presented as functions of curing temperatures in Figure 4b. It can be seen that increasing curing temperature, the conversion rates α increased correspondingly. And increasing heat rates, the conversion rate α varied at the same curing temperature, indicating heat rates would obviously affect the polymerization processes.

Based on the iso-conversion principle, temperature dependent of iso-conversion rate can be utilized to calculate the activation energy. Among the various empirical equations, Straink proposed somewhat more accurate estimates of E_α_ are accomplished when setting B = 1.92 and C = 1.0008, so that the equation turns into Equation (2):(2)$$ln\left(\frac{\beta_{i}}{T_{\alpha ,i}^{B}}\right)=Const-C\left(\frac{E_{\alpha }}{RT_{\alpha }}\right)$$

For the equation, a plot of $\ln\beta_{i}/T_{\alpha ,i}^{1.92}$ vs. $1/T_{\alpha }$ value at the same fractional extent of conversion from a series of dynamic DSC experiment at different heating rates would result in a straight line with a slope of $-1.0008E_{\alpha }/R$. The plots of $\ln\beta_{i}/T_{\alpha ,i}^{1.92}$ as a function of $1/T_{\alpha }$ value for α = 0.1–0.9 are shown in Figure 5a. A good linear relationship is observed from the plots shown in Figure 5a. Repeating the procedure, the E_α_ values corresponding to different α from DSC curing curves can be obtained and shown in Figure 5b. From the plots, the values of apparent activation energies (E_α_) were shown to be somewhat dependent of the curing extent. It was obvious that E_α_ was independent of the curing extent in the range from 0.1 to 0.6, while in the range from 0.7 to 0.9, E_α_ reduced significantly with the increasing of curing extent. It can be ascribed to the piecewise polymerization of FAEN-Bz resin. When the curing extent was low, polymerization of FAEN-Bz was dominated by chemical polymerization driven by thermos-motive, and when the curing extent increased, polymerization translated to diffusion polymerizations, which were dominated by the motion of molecular chains. Thus, the average value of E_α_ in the chemical polymerization can be obtained in the range from α = 0.1–0.6, which was about 127 kJ/mol.

Moreover, in comparison with the E_α_ obtained from gelation and thermodynamic analysis, the average values were different. It may contribute to the different ways of collecting data. For the gelation method, the analysis based on gelation time reflected the general polymerizations. In the analysis of thermodynamic data from the results of the DSC, more details on the curing behaviors were taken into consideration. Thus, thermodynamic analysis based on DSC may be more accurate in the discussion of polymerization mechanisms.

### 3.3. Curing Behaviors of FAEN-Bz and Its Polymerization Mechanisms

DSC was used to characterize the curing behaviors of FAEN-Bz at different temperatures, shown in Figure 6a. One obvious exothermic peak was obviously for FAEN-Bz, which can be assigned to ring-opening polymerization of oxazine rings [12,36]. Elevating the curing temperature, exothermic peaks shifted to a higher temperature range, indicating that the polymerization of oxazine rings was impressible to the heat treatments. As presented, the exothermic peak of FAEN-Bz treated at 180 °C was obviously higher than that treated at 160 °C. As we know, the polymerization of self-catalyzed resins were dominated by the ‘initiator’. For FAEN-Bz, the initiator of its polymerization was the trace of active hydrogen remained in the resins. Thus, at a lower curing temperature, trace hydrogen catalyzed the polymerization slowly and active hydroxyl groups generated from the ring-opening reaction of oxazine rings. When the curing temperature increased, the generated hydroxyl groups catalyzed the further polymerization of oxazine rings sharply [34]. Increasing the curing temperature persistently, no obvious exothermic peaks can be observed in the range from 200 to 250 °C for the poly(FAEN-Bz), shown in Figure 6a. According to the polymerization of pristine benzoxazine rings, ring-opening polymerization occurred at around 230 °C and finished at about 260 °C. Thus, post-cured treatment in this work was designed to obtain the completely cured poly(FAEN-Bz). Figure 6b showed the thermal properties of poly(FAEN-Bz) cured at various temperatures, and obvious transformations can be observed. With continue increasing the curing temperatures, transformations occurred in higher temperature range. The *tan δ* of dynamic mechanical analysis (DMA) of glass fiber feinforced FAEN-Bz composites (FAEN-Bz/GF) cured at 200 °C for 4 h and 240 °C for 2 h was presented in Appendix A. Glass transition temperature (*T*_g_) was about 233 °C.

To further confirm the structural transformation during post-cured processes, FTIR spectra were presented in Figure 7. In the spectra, characteristic absorption peaks/bands of ring-opened oxazine rings and nitrile groups were marked. Obviously, absorption peak appeared at about 2230 cm^−1^ had no changes after being treated at various temperatures, which was ascribed to nitrile groups, indicating that nitrile groups were not involved in the polymerization during the temperature from 240 to 280 °C [17,37]. In the previous work, nitrile groups participated into the polymerizations and ring-forming polymerization occurred along with the ring-opening polymerization of oxazine rings at about 240 to 280 °C [36,38]. The differences can be attributed to the phthalonitrile and single nitrile, which showed distinct reactivity. During the polymerization of phthalonitrile, triazine and phthalocyanine rings can be formed logically. For the single nitrile groups in FAEN-Bz, the ring-forming polymerization was much difficult due to the low nitrile content and high steric hindrance. The characteristic absorption observed at around 3289–3555 cm^−1^ can be mainly assigned to the intermolecular and intramolecular H-bonds, which were formed between the hydroxyl and nitrogen atoms generated from the ring-opening polymerization of oxazine rings [39,40]. As the references reported that the intramolecular H-bonds could turn into intermolecular H-bonds when the temperature went on which contributed to the increase of surfce free energy of cured matrix [41,42]. It was well known that the formation of hydrogen bonds was sensitive to temperature. The intermolecular formation of hydrogen bonds would be reduced at elevated temperature if the H-bonds were not sufficiently strong. It can be seen that with increasing the treatment temperature, the absorption intensity of H-bonds slightly decreased, indicating that the H-bonds were not stable at the temperature above 240 °C [40].

In order to study the curing reaction of FAEN-Bz, the structure changes as a function of temperature was characterized by in situ FTIR ranging from 50 to 300 °C. The results are shown in Figure 8. The characteristic asorbtion at 953, 1200, 2230 and 3400 cm^−1^ were corresponed to the antisymmetric oxazine ring stretch, –C–F, intermolecular and intramolecular association of hydroxyl bonds, respectively. It can be seen that the absorption at 953 and 3400 cm^−1^ gradually decreased with the increasing of temperature. Before the temperature 175 °C, the ring-opening reaction of oxazine rings occured slowly and the H-bonds would be increased with the hydroxyl group generated from the ring-opening reaction of oxazine rings. When the temperature went on (>250 °C), due to the higher temperature the reaction of oxazine rings intensively carried out which leaded to the absorption of oxazine ring rapidly decreased. With the ring-opening of oxazine rings, the formation of H-bonds between intramolecular and intermolecular chains increased [42]. However, the characteristic asorbtion of nitrile group at 2230 cm^−1^ did not exhibit an obvious decrease at various temperature except for 300 °C. This also indicated that curing reaction of nitrile groups was not involved in the polymerization of FAEN-Bz.

In sum, the polymerization processes of FAEN-Bz can be summarized as follows and shown in Scheme 2: ring-opening polymerization occurred at about 160 °C in the presence of traces of hydroxyl, and at that temperature the polymerization happened slowly (Scheme 2a); then, elevating the curing temperature to about 180 °C, the ring-opening polymerization of oxazine rings speeded up (Scheme 2b). Amounts of hydroxyl groups were generated and H-bonds formed between nitrogen atoms and hydrogen (Scheme 2c). With continuously increasing the curing temperatures, the polymerization got slower gradually, due to the hindrance of molecular chains. When the temperature was 240 °C and even higher, with the ring-opening of oxazine rings, the formation of H-bonds between intramolecular and intermolecular chains increased. Thus, the possible structures of final polymers (heat treated at 280 °C) can be simulated as Scheme 2d.

### 3.4. Thermal Stability of Poly(FAEN-Bz) and the Thermal Decomposition Mechanism

Thermal properties of poly(FAEN-Bz) polymer was also examined under nitrogen atmosphere with the heating rate of 20 °C/min from 50 to 800 °C. The curves were shown in Figure 9 and the main data was collected in Table 2, including the weight loss of 5 wt % (*T*_d5%_), 10 wt % (*T*_d10%_) and the char yield at 800 °C. According to the TGA curves and the data listed in Table 2, it was obvious that the decomposition temperature of *T*_d5%_ and char yield were greatly affected by the treatment temperature and increased with increasing curing temperatures. The poly(FAEN-Bz) cured at 280 °C exhibited thermal stability of *T*_d5%_ and *T*_d10%_ up to 407 and 449 °C, and the char yield at 800 °C was 63.5% under nitrogen. Compared to thermal properties reported by Zeng (*T*_d5%_ ~356.7 °C, char yield at 800 °C ~39.9%) [43], Wang (*T*_d5%_ ~363.7 °C, char yield at 800 °C ~44%) [13] and Khan (*T*_d10%_ ~360 °C, char yield at 800 °C ~3.3%) [44], poly(FAEN-Bz) exhibited a excellent thermal stability. Moreover, the curves of thermal decomposition showed the same trend, indicating that the decomposition processes were consistent. Figure 9b presented the DTG curves of poly(FAEN-Bz), which represented the temperature at which the materials decomposed at the maximum rate. We could see intuitively that all of poly(FAEN-Bz) showed double decomposition processes in Figure 9b. The double maximum decomposition peaks of poly(FAEN-Bz) appeared at 400 and 540 °C, respectively. Moreover, the maximum decomposition temperatures did not change with increasing the curing temperatures, indicating the decomposition of poly(FAEN-Bz) was dominated by the intimate structures, and increasing the crosslinking degree cannot intensively improve the thermal stability of poly(FAEN-Bz) in this cured condition.

Considering the polymerization mechanisms and the intimate structures of poly(FAEN-Bz), the thermal decomposition processes can be concluded as follows (shown in Scheme 3): firstly, the Mannich bridge was broken and the long molecular structures of poly(FAEN-Bz) turn into aromatic ester nitrile segments, which possessed outstanding thermal stability (Scheme 3a) [9]. At the elevated temperatures, the segments began to collapse along with the fracture of chemical bonds (Scheme 3b) [27].

To further verify the thermal stability of poly(FAEN-Bz), the integral program integral decomposition temperature (IPDT) was used and described in Equation (3) [45].
(3)$$\mathrm{IPDT}=A^{*}K^{*}\times \left(T_{f}-T_{i}\right)+T_{i}$$where A was the area ratio of the total experiment curve defined by the total TGA thermogram curve. T_i_ was the initial temperature and T_f_ was the final temperature. In this study, T_i_ and T_f_ were 50 and 800 °C, respectively. A and K can be calculated by Equations (2) and (3). The values of S1, S2 and S3 are determined by Figure 10.
(4)$$A*=\frac{S1+S2}{S1+S2+S3}$$
(5)$$K*=\frac{S1+S2}{S1}$$

The IPDT values of poly(FAEN-Bz) cured at various temperatures were presented in Table 3. It was obvious that the thermal stability increased with increasing the curing temperatures. The IPDT showed an upward trend until 2203 °C, which can be attributed to the complete polymerization of oxazine rings and the high thermal stability of the aromatic ester molecular segments. Rich aromatic nucleus and heterocyclic rings were in favor of enhancing the inherent thermal stability of the polymers.

### 3.5. Phase Morphology of the Poly(FAEN-Bz) Polymer

SEM images of poly(FAEN-Bz) heat-treated at 260 °C and 280 °C were presented in Figure 11a,b, respectively. The images showed the evolution of the phase morphology as the curing temperature increased. After being treated at 260 °C, the ring-opening polymerization of oxazine rings resulted to a compared homogeneous phase (Figure 11a). It was obvious that slight phase inversion coexistence in sight and spherical nodules are visible. When the curing temperature was 280 °C, the ring-opening polymerization of oxazine rings can be almost accomplished, which can be confirmed with the results of DSC and TGA. Thus, the image in Figure 11b showed a well homogeneous phase, which was mainly made of Mannich bridge and aromatic ester nitrile segments in an orderly arrangement. Figure 11c,d showed the fracture images of poly(FAEN-Bz) after being extracted with Soxhlet extraction, and 1-methyl-2-pyrrolidinone (NMP) was used as extracting solvent. The incomplete crosslinked resin would be dissolved and the remainder can be used to evaluate the crosslinking degrees. In Figure 11c, the extracted fracture image showed a homogeneous phase and many pinholes appeared, which can be attributed to the dissolution of uncured molecular that was restricted in the crosslinking network. However, in Figure 11d, no obvious changes can be observed in compared with that shown in Figure 11b. It can be assigned to the almost accomplished polymerization of oxazine rings in the polymer. According to the extracted results, curing degrees of poly(FAEN-Bz) cured at 260 and 280 °C were calculated, and the degrees were 85.0% and 96.3%, respectively, indicating relatively high crosslinking degrees.

SAXS was applied to investigate the phase morphology of FAEN-Bz cured at 260 °C for 2 h and 280 °C for 2 h and the results were shown in Figure 12. Figure 12a,b were corresponded to the FAEN-Bz cured at 260 and 280 °C/2 h, respectively. As shown in Figure 1, no characteristic diffraction peaks were observed [46]. This indicated that the homogeneous phase of cured FAEN-Bz.

### 3.6. Surface Wettability of Poly(FAEN-Bz)

The surface charge and its wettability determine the surface properties and subsequently the dielectric properties. Contact angle measurements of wettability are essential evaluation of surface charge. As was well known, polymer surfaces with a high content of –CH_3_, –CH=CH_2_ groups form hydrophobic surfaces (ɵ_a_ ≥ 80°), while –COOH, –NH_2_ groups form moderately hydrophobic surfaces (ɵ_a_ = 48–62°) and -OH groups form hydrophilic surfaces (ɵ_a_ ≤ 35°) [47]. In this work, smooth and clean samples were prepared and then water was dripped on the solid surface to evaluate the contact angles, shown in Figure 13. The surface energy was calculated by Fowkes method and ZHU method, as shown in Table 4. All of the values of contact angle were ranging from 76.9° to 87.4°, indicating the moderately hydrophilic surfaces of poly(FAEN-Bz). With increasing the curing temperatures, the values of contact angle increased, which indicated that the hydrophilic characterization enhanced and the hydrophobic characterization increased. It can be attributed to the fact that with the ring-opening of oxazine rings, the formation of H-bonds between intramolecular and intermolecular chains increased. This trends were well consistent with the published literatures [41,42].

### 3.7. Dielectric Properties of Poly(FAEN-Bz) Cured at Various Temperatures

The variations of dielectric constant and dielectric loss with regard to frequency for poly(FAEN-Bz) cured at various temperatures were presented in Figure 14. Dielectric properties for polymers are mainly influenced by relaxation mechanisms and dielectric polarization in bulk of the composites [48]. It can be observed from Figure 14a that dielectric constant was frequency dependent for poly(FAEN-Bz) polymers and decreased with increasing the frequency. It’s reported that dielectric constant would continuously decrease at high frequency due to the fact that orientable dipolar groups and relaxation of molecular chains could not keep pace with that of the alternating field. However, with increasing the cured temperatures, dielectric constant of poly(FAEN-Bz) slightly increased. Figure 14a showed that poly(FAEN-Bz) cured at 280 °C possessed the highest dielectric constants (~5.0), which can be assigned to the polar hydroxyl existed in the matrix resin. The increasing of dielectric constant can be attributed to the two facts: (1) with the curing temperature went on the intramolecular H-bonds would turn into intermolecular H-bonds to increase the surface energy thus to increase the dielectric constants; (2) the H-bonds could be destroyed under high temperature to form free –OH to increase the dielectric constants [40,41]. In Figure 14b, low dielectric loss was presented and it’s obvious that dielectric loss was a frequency-independence parameter for poly(FAEN-Bz) cured at 240 and 260 °C, while for poly(FAEN-Bz) cured at 280 °C, the dielectric loss was frequency dependent. For poly(FAEN-Bz) with polar hydroxyl, conduction loss and relaxation loss contributed most to the low dielectric loss, with respect to frequency. In high frequency of 500 Hz–1 MHz, an increasing of dielectric loss appeared, that can be attributed to the significant relaxation polarization loss generated from orientable dipoles in molecular groups attached perpendicular to longitudinal polymer chains. Then, at higher frequency, the conduction loss dominated dielectric loss for polymers, showing a slight decrease in curves shown in Figure 14b. In sum, dielectric properties of poly(FAEN-Bz) can be tuned by controlling the crosslinking degrees.

## 4. Conclusions

A self-polymerization benzoxazine resin with aromatic ester nitrile segments was successfully synthesized by solution reaction. Structures were verified by ^1^H-NMR, FTIR and in situ FTIR spectroscopy. Curing kinetics of the molecular was discussed with macromolecular gel method and DSC analysis. Various activation energy of the molecular were obtained. Combining the structural transformation shown in FTIR spectra, the curing behaviors and polymerization processes were proposed. Then, fracture surface images, SAXS and crosslinking degrees of the polymers were used to confirm the polymerization processes. Surface wettability and dielectric properties of various polymers were also investigated to confirm the polymerization processes and results indicated that ring-opening polymerization of oxazine rings dominated the polymerization and H-bonds existed in inter/intramolecular. Also, thermal decomposition of the polymers were also investigated and results indicated that the polymers processed outstanding thermal stability. In summary, controllable dielectric properties with good thermal stability, which can be tuned by controlling the curing temperatures and time, poly(FAEN-Bz) was believed to be candidates for application of printed circuit substrate materials.

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
