# Peer review of "Benzoxazine Containing Fluorinated Aromatic Ether Nitrile Linkage: Preparation, Curing Kinetics and Dielectric Properties"

_polymers, 2019, doi:10.3390/polym11061036_

Round 1

Reviewer 1 Report

The author submitted “Benzoxazine containing fluorinated aromatic ether nitrile linkage: Preparation, curing kinetics and dielectric properties” to publish in Polymers. In this manuscript, authors describe synthesis the modified benzoxazine containing fluorinated aromatic ether nitrile linkage. Then, its properties including thermal stability, surface wettability and dielectric properties were analyzed to discuss the influence of the fluorinated aromatic ether nitrile linkage on the properties of benzoxazine-based polymers. I think that this work is interesting, and the novelty of this work is good enough to publish in this journal. Therefore, this manuscript should be published after Major revisions. Here, my comments:

1. The introduction should be improved and authors should cite some new references in this field.

2. there is wrong in page 4, line 129, the resonance peak appearing at 3.76 ppm was assigned to the hydroxyl protons of amino groups. Also, resonance appearing at 6.82 and 7.46 ppm were aromatic protons with nitrile groups.

3. The purity of FAEN-NH2 and FAEN-BZ is quite low, according to 1H-NMR. The authors should do 1H-NMR measurement with a high pure samples. Also, 13C-NMR is required.

4. Authors should provide DMA results.

5. Authors should do FT-IR measurement of FAEN-BZ at various temperatures and combine this figure with DSC results.

6. Authors should cite this reference: (a) doi.org/10.1002/macp.201800306 and (b) 10.1039/C7PY01026E. 

Author Response

Reviewer 1: The author submitted “Benzoxazine containing fluorinated aromatic ether nitrile linkage: Preparation, curing kinetics and dielectric properties” to publish in Polymers. In this manuscript, authors describe synthesis the modified benzoxazine containing fluorinated aromatic ether nitrile linkage. Then, its properties including thermal stability, surface wettability and dielectric properties were analyzed to discuss the influence of the fluorinated aromatic ether nitrile linkage on the properties of benzoxazine-based polymers. I think that this work is interesting, and the novelty of this work is good enough to publish in this journal. Therefore, this manuscript should be published after Major revisions. Here, my comments: 1. The introduction should be improved and authors should cite some new references in this field. Answer to reviewer: Thanks for your comments and suggestions. The Introduction section was carefully modified. The references in manuscript was partly replaced and the new references about the study of benzoxazine resin and modification on toughness of benzoxazine resin, introduction of polymer poly(arylene ether nitrile) and some other relate references were added in revised manuscript. The modifications in revised manuscript were marked as Red. 2. There is wrong in page 4, line 129, the resonance peak appearing at 3.76 ppm was assigned to the hydroxyl protons of amino groups. Also, resonance appearing at 6.82 and 7.46 ppm were aromatic protons with nitrile groups. Answer to reviewer: We are sorry for the mistakes here. Authors corrected the wrong writing in revised manuscript and marked as red. The revised writing was that the resonance peak appearing at 3.76 ppm was assigned to the protons of amino groups. Resonance appearing at 6.82 and 7.46 ppm were protons protons from aromatic rings with nitrile groups. 3. The purity of FAEN-NH2 and FAEN-BZ is quite low, according to 1H-NMR. The authors should do 1H-NMR measurement with a high pure samples. Also, 13C-NMR is required. Answer to reviewer: Thanks for your comments and suggestions. In this work, amino terminated fluorinated aromatic ether nitrile linkage was synthesized and characterized by 1H-NMR. Because of the relative low content of amino groups in the molecular chains, the intensity of H of amino groups was low. Moreover, due to the similar solubility and polarity, by-product of FAEN-NH2 was difficult to remove completely. In this work, 1H-NMR and FTIR spectra were applied to confirm the structures of FAEN-NH2 and FAEN-Bz, and results indicated that the designed products were obtained. So, 13C-NMR spectra were not presented in this manuscript. 4. Authors should provide DMA results. Answer to reviewer: According to your comments, the DMA (Q800, TA Instruments, USA) tests were supplemented. The DMA was tested by the glass fiber (GF, EW170-100, Shenyang Gaote glass fiber Co., Ltd., Shenyang, China.) reinforced FAEN-Bz composite laminates. The FAEN-Bz/GF composites prepared as follows. 8 layers of GF were impregnated with FAEN-Bz solution. Then, the prepregs dried in oven at 80 oC for 20 min and 160 oC for 15 min to remove the solvent, and the dried prepregs were placed in stainless mold under 20 MPa at 200 oC for 4 h and 240 oC for 2 h to prepare FAEN-Bz/GF composites. DMA (three-point bending) was tested from 50 oC to 260 oC (5 oC/min) under air atmosphere with the frequency of 1 Hz. The results were presented as follows. According to the glass transition theory, the temperature at the maximum peak of tan δ curve can be defined as glass transition temperature Tg. The Tg of FAEN-Bz/GF composites cured at 200 oC for 4 h and 240 oC for 2 h exhibited a high Tg (about 233 oC). Compared to the polybenzoxazine previously reported [1], polyFAEN-Bz possessed good thermal stabilities. Figure 1. tan δ of DMA tests. FAEN-Bz/GF composites with curing procedure 200 oC/4 h and 240 oC/2 h. References: 1. Kobzar, Yaroslav L., Ihor M. Tkachenko, Valery N. Bliznyuk, and Valery V. Shevchenko. "Fluorinated Polybenzoxazines as Advanced Phenolic Resins for Leading-Edge Applications." Reactive & Functional Polymers 133 (2018): 71-92. 5. Authors should do FT-IR measurement of FAEN-BZ at various temperatures and combine this figure with DSC results. Answer to reviewer: Thanks and your suggestion was adopted in the revised manuscript. The in-situ FT-IR measurement of FAEN-Bz was characterized ranging from 50 to 300 oC. The results were shown in Figure 1 (Figure 8 in revised manuscript). The description in revised manuscript was copied as follows. In order to study the curing reaction of FAEN-Bz, the structure changes as a function of temperature was characterized by in situ FTIR ranging from 50 oC to 300 oC. The results were shown in Figure 8. The characteristic asorbtion at 953, 1200, 2230 and 3400 cm-1 were corresponed to the antisymmetric oxazine ring stretch, -C-F, intermolecular association of hydroxyl bonds, respectively. It can be seen that the absorption at 953 and 3400 cm-1 gradually decreased with the increasing of temperature. Before the temperature 175 oC, the ring-opening reaction of oxazine rings occured slowly and the H-bonds interaction would be increased with the hydroxyl group generated from the ring-opening reaction of oxazine rings. When the temperature went on (> 250 oC), due to the higher temperature the reaction of oxazine rings intensively carried out which leaded to the absorption of oxazine ring rapidly decreased. At the same time, the H-bonds would be destroyed under the high temperature. Thus, the absorption at around 3400 cm-1 also rapidly decreased. However, the characteristic asorbtion of nitrile group at 2230 cm-1 did not exhibit an obvious decrease at various temperature except for 300 oC. This also indicated that curing reaction of nitrile groups was not involved in the polymerization of FAEN-Bz. Figrue 8. In situ FTIR spectra of FAEN-Bz from 50 oC to 300 oC. 6. Authors should cite this reference: (a) doi.org/10.1002/macp.201800306 and (b) 10.1039/C7PY01026E. Answer to reviewer: Thanks for your suggestions. The references were cited in the revised manuscript.

Reviewer 2 Report

The paper is well written, few words to possible aplications would be good. I would also add new related to this work Papers Polymers 2019, 11(4), 679; https://doi.org/10.3390/polym11040679 One-Pot Synthesis of Amide-Functional Main-Chain Polybenzoxazine Precursors Canan Durukan et al. and Synthesis and Crosslinking of Polyether-Based Main Chain Benzoxazine Polymers and Their Gas Separation Performance Muntazim Munir Khan et al Polymers 2018, 10(2), 221, online 23 February 2018, open access

Author Response

Reviewer 2: The paper is well written, few words to possible applications would be good. I would also add new related to this work Papers Polymers 2019, 11(4), 679; https://doi.org/10.3390/polym11040679 One-Pot Synthesis of Amide-Functional Main-Chain Polybenzoxazine Precursors Canan Durukan et al. and Synthesis and Crosslinking of Polyether-Based Main Chain Benzoxazine Polymers and Their Gas Separation Performance Muntazim Munir Khan et al Polymers 2018, 10(2), 221, online 23 February 2018, open access Answer to reviewer: Thanks for your comments and suggestions on this work. The references were cited in the revised manuscript. The writing, words and grammar were carefully checked and modified. The modification in revised manuscript was marked as red.

Reviewer 3 Report

Generaly, the materials studied and presented results in this paper are very interesting, valueable and useful from industrial point of view. Some drawbacks are mentioned below.

scheme 1: I have doubts about the first reaction. Could you please write more details about conditions of this reaction and cite some literature about it?

line 132: "Similarly, the aromatic protons with nitrile groups were observed at 6.82 and 7.46 ppm, indicating that the designed structures were obtained." - It will be useful to show also NMR for compound a from scheme 1. In this sentance the statement "the aromatic protons with nitrile groups" is misleading. I understand that it means protons from aromatic rings with nitrile groups. Please, correct.

line 152: "It can be seen that with increasing test temperatures, the gel time significantly decreased, indicating that the polymerization of FAEN-Bz was sensitive to 153C, gel time of FAEN-Bz was less than 3 temperatures. Moreover, when the temperature was 190 mins, which had direct meaning to the preparation of FAEN-Bz composites." - This resins are thermosetting, thus, this observation is rather obvious. It would be better to focus on comparison between tested materials.

line 251: "the hydrogen-bonding reaction" - It sounds not accurate. For me it is simply formation of hydrogen bonds. Please explain what do you mean by hydrogen-bonding reaction?

Scheme 2: The hydrogen bonds are presented on this scheme as between O and O as well as O and N that is not true. Please correct and write it preceisely.

Thermal stability: it would be interesting to compere thermal stability of FAEN-Bz with other resins.

Scheme 3. Too small evidance are presented to prove this hypothetical decomposition process. I know it is only the proposition but there are some thechniques to prove it, even in small way,e.g. py-GCMS. I realize that the Authors can not have possibility to perform such studies but maybe they can try other techniques or support this proposed decomposition reaction by literature.

In my opinion, SEM is not appropaite to discuss phase morphology. In my opinion SAXS/WAXS technique can be useful.

Table 4. Contact angle measurements must have the error limits. It is very important as the presented values of the contact angle might be in the error limit and not important statisticaly.

Author Response

Reviewer 3: Generaly, the materials studied and presented results in this paper are very interesting, valueable and useful from industrial point of view. Some drawbacks are mentioned below. 1. Scheme 1: I have doubts about the first reaction. Could you please write more details about conditions of this reaction and cite some literature about it? Answer to reviewer: Thanks for your comments on our work. We are sorry for the incomplete conditions in the first reaction. We have added the more conditions in section 2.2 and the Scheme 1 was modified. The modified Scheme 1 was copied as follows. Scheme 1. The synthetic route of designed FAEN-Bz: (a) chlorinated terminated monomers (FAEN-Cl), (b) amino terminated monomers (FAEN-NH2) and (c) benzoxazine containing fluorinated aromatic ether nitrile (FAEN-Bz). 2. line 132: "Similarly, the aromatic protons with nitrile groups were observed at 6.82 and 7.46 ppm, indicating that the designed structures were obtained." - It will be useful to show also NMR for compound from scheme 1. In this sentance the statement "the aromatic protons with nitrile groups" is misleading. I understand that it means protons from aromatic rings with nitrile groups. Please, correct. Answer to reviewer: Thanks for your comments. The structure of FAEN-NH2 and FAEN-Bz were characterized by 1H-NMR and the results showed that the monomers were successfully prepared. In our previous researches, the synthetic method in Scheme 1(a) was applied to synthesize the poly(arylene ether nitrile) which was a kind of high performance thermoplastic polymer. Thus, the structure of Scheme 1(a) could be synthesized via the method. The references about preparation of poly(arylene ether nitrile) and synthesis of aromatic ether nitrile linkage was added in revised manuscript and copied as follows [1, 2]. According to your suggestion, sentance the statement "the aromatic protons with nitrile groups" was revised as "protons from aromatic rings with nitrile groups". 3. line 152: "It can be seen that with increasing test temperatures, the gel time significantly decreased, indicating that the polymerization of FAEN-Bz was sensitive to temperature, gel time of FAEN-Bz was less than 3 min. Moreover, when the temperature was 190 oC, which had direct meaning to the preparation of FAEN-Bz composites." - This resins are thermosetting, thus, this observation is rather obvious. It would be better to focus on comparison between tested materials. Answer to reviewer: Thanks for your suggestions. In the section 3.2.1, the gel time of FEAEN-Bz was measured at different temperature. According to the gel time, the reaction activity (Eα) was calculated via Arrhenius equation. As the results showed that, the average values (Eα) of FAEN-Bz was calculated as 98.7 kJ/mol, which was slightly lower than that of benzoxazine (102-116 kJ/mol) reported. This can be attributed to the fact that electron withdrawing groups (nitrile groups and fluorine atoms) have induced the π electron cloud of benzene slightly shifted to O and N atoms in oxazine rings, which accelerated the polymerization of FAEN-Bz. Thus, the value of Eα was slightly lower than the benzoxazine previously reported. In the practical operation, a proper gel time (3-5 min) is needed to prepare the FAEN-Bz polymer or their composites. When the temperature was 190 oC, the gel time was about 3 min and can be chosen as the initial processing temperature. Following your comments, the description “indicating that the polymerization of FAEN-Bz was sensitive to temperatures” was removed. 4. line 251: "the hydrogen-bonding reaction" - It sounds not accurate. For me it is simply formation of hydrogen bonds. Please explain what do you mean by hydrogen-bonding reaction? Answer to reviewer: We are sorry for making an inaccurate explanation here. The meaning of the description "the hydrogen-bonding reaction" would like to explain the formation of hydrogen bonds between H, O and N. But, the description "the hydrogen-bonding reaction" was not accurate in manuscript. The "the hydrogen-bonding reaction" was modified as "formation of hydrogen bonds ". 5. Scheme 2: The hydrogen bonds are presented on this scheme as between O and O as well as O and N that is not true. Please correct and write it preceisely. Answer to reviewer: Thanks for your suggestions on the modification of Scheme 2. The wrong description in Scheme 2 was corrected and copied as follows. Scheme 2. Possible polymerization processes of FAEN-Bz: (a) the initiation of ring-opening polymerization, (b) the polymerization of oxazine rings at 180 oC, (c) the formation of H-bond inter/intra molecular and (d) the possible structure of final polymers. 6. Thermal stability: it would be interesting to compere thermal stability of FAEN-Bz with other resins. Answer to reviewer: The thermal stability of other benzoxazine resins reported by Zeng (Td5% ~ 356.7 oC, char yield at 800 oC ~ 39.9%) [3]. Wang (Td5% ~ 363.7 oC, char yield at 800 oC ~44%) [4]. and Khan (Td10% ~360 oC, char yield at 800 oC~3.3%) [5] were added in revised manuscript. The poly(FAEN-Bz) resin exhibited an excellent thermal stability. 7. Scheme 3. Too small evidance are presented to prove this hypothetical decomposition process. I know it is only the proposition but there are some thechniques to prove it, even in small way, e.g. py-GCMS. I realize that the Authors can not have possibility to perform such studies but maybe they can try other techniques or support this proposed decomposition reaction by literature. Answer to reviewer: Thanks for your comments and suggestions on possible decomposition process in Scheme 3. According to the references and the previous researches on Poly(arylene ether nitrile) in our laboratory, the two maximum decomposition peaks at 400 oC and 540 oC were corresponded the decomposition of Mannich bridge structure and aromatic ester main chains [2, 6]. 8. In my opinion, SEM is not appropaite to discuss phase morphology. In my opinion SAXS/WAXS technique can be useful. Answer to reviewer: According to the comments, the small angle X-ray scattering (SAXS, Bruker AXS D8, Germany) was applied to investigate the phase morphology of FAEN-Bz cured at 260 oC/2 h and 280 oC/ 2 h. The results were presented as follows Figure 1 (Figure 12 in revised manuscript). Figure 1(a) and Figure 1(b) were corresponded to the FAEN-Bz cured at 260 oC/2 h and 280 oC/ 2 h, respectively. As shown in Figure 1, no characteristic diffraction peaks were observed. This confirmed that the homogeneous phase of cured FAEN-Bz. Figure 1. SAXS of poly(FAEN-Bz) cured at various temperatures: (a) 260 oC-2 h; (b) 280 oC-2 h. 9. Table 4. Contact angle measurements must have the error limits. It is very important as the presented values of the contact angle might be in the error limit and not important statisticaly. Answer to reviewer: Thanks for your suggestions. The error limits were added in Table 4 and copied as follows. Table 4. Contact angle and surface energy of poly(FAEN-Bz) Sample Liquid medium Surface tension of liquid phase media [(mN/m)] Contact angle (º) Calculation method Solid surface energy [(J/m2)] 240oC water 72.800003 76.9±0.8 Fowkes method 0.02738 260oC water 72.800003 81.7±0.3 Fowkes method 0.02383 280oC water 72.800003 87.4±2.3 Fowkes method 0.01989 240oC water 72.800003 76.9±0.8 ZHU method 0.05906 260oC water 72.800003 81.7±0.3 ZHU method 0.05646 280oC water 72.800003 87.4±2.3 ZHU method 0.05310 References: 1. Du, Ronghua, Wenting Li, and Xiaobo Liu. "Synthesis and Thermal Properties of Bisphthalonitriles Containing Aromatic Ether Nitrile Linkages." Polymer Degradation And Stability 94, no. 12 (2009): 2178-83. 2. Tang, Hailong, Jian Yang, Jiachun Zhong, Rui Zhao, and Xiaobo Liu. "Synthesis and Dielectric Properties of Polyarylene Ether Nitriles with High Thermal Stability and High Mechanical Strength." Materials Letters 65, no. 17-18 (2011): 2758-61. 3. Zeng, Ka, Jiayue Huang, Junwen Ren, and Qichao Ran. "Curing Reaction of Benzoxazine under High Pressure and the Effect on Thermal Resistance of Polybenzoxazine." Macromolecular Chemistry And Physics 220, no. 1 (2019). 4. Wang, Xin, Lishuai Zong, Jianhua Han, Jinyan Wang, Cheng Liu, and Xigao Jian. "Toughening and Reinforcing of Benzoxazine Resins Using a New Hyperbranched Polyether Epoxy as a Non-Phase-Separation Modifier." Polymer 121 (2017): 217-27. 5. Khan, Muntazim Munir, Karabi Halder, Sergey Shishatskiy, and Volkan Filiz. "Synthesis and Crosslinking of Polyether-Based Main Chain Benzoxazine Polymers and Their Gas Separation Performance." Polymers 10, no. 2 (2018). 6. Rishwana, S. Shamim, A. Mahendran, and C. T. Vijayakumar. "Studies on Structurally Different Benzoxazines: Curing Characteristics and Thermal Degradation Aspects." High Performance Polymers 27, no. 7 (2015): 802-12.

Reviewer 4 Report

 Benzoxazine containing fluorinated aromatic ether nitrile linkage: Preparation, curing kinetics and dielectric properties

The manuscript describes the preparation of benzoxazine containing fluorinated aromatic ether nitrile linkage. Curing behaviors and curing kinetics of designed monomers was also studied. The obtained polybenzoxazine exhibits an excellent material and the results are very promising. the manuscript losses the scientific English writing style. Therefore, this manuscript is not yet ready to be published in this journal. There are questions/suggestions/corrections that the authors should consider before publication:

1. Lines 28 – 31: “With the development of science and technology, thermosetting resins play an important role in electronics, substrate materials, aeronautics-aerospace industry and new energy applications, due to the advantages including high specific strength and compressive strength, corrosion resistance, good mechanical and electrical properties, and outstanding bonding properties[1-4].” This sentence is very long; therefore, it should be rewritten in simple sentences.

2. Lines 55 and 56: The abbreviations should be removed.

3. Line 60: “Recent years, …” should be corrected to read “in recent years, …” or “Recently,….”

4. All references are combined with previous words; therefore a space should be added before the references. For example, Line 31 “properties[1-4].”

5. Line 86: “our work reported previously with minor modifications” the authors should avoid using pronouns. Also, this sentence should be rewritten in corrected English grammar.

6. Lines 89 and 90: “Then 3-Aminophenol was added into the solution, and amino terminated monomers (FAEN-NH2) were obtained, shown in scheme 1 (b).” This sentence should be rewritten in corrected English grammar.

7. Lines 96 and 97: “Finally, the resulted mixture was dried in oven at 110 oC 96 for 4 h to remove the solvents to obtain the FAEN-Bz[10].” The authors should recheck the English language grammar.

8. Lines 106-108: “the viscosity solution was cast on a clean glass plate in an oven to evaporate the solvent with the procedure of…….” The authors should improve the English language of the sentence.

9.  

10.       Line 115: “Differential scanning calorimetric (DSC) investigated ….” is not complete sentence.

11.       Line 140: “Characteristic absorption band of -NH2 was observed at 3381 cm-1.” However, the band 3381 cm-1 was assigned for intermolecular interaction as shown in Lines 142-144 “a wide absorption band at around 3381 cm-1 was observed, which can be assigned to the intermolecular association of hydroxyl bonds, generated from the trace hydrogen and nitrogen atoms in benzoxazine rings.” The authors should clarify the both sentence for assigned groups for the band at 3381 cm-1.

12.       Line 140, A comma should be added before the word “respectively”.

13.       Line 190: “….when setting B = 1.92 and C = 1.0008”. Because the constants B and C were given; therefore, Eq. 2 should contain these constants instead of the numbers.

14.       Line 285 and 286: “The double maximum decomposition peaks of poly(FAEN-Bz) appeared at 400 oC and 540 oC, respectively.” Why this polymer shows double maximum decomposition peaks? The authors should judge their investigations.

15.       Lines 288 and 289: “ …. increasing the crosslinking degree cannot improve the thermal stability obviously” I disagree with this statement because the previous investigations reported that crosslinking enhances the thermal stability of polymers, especially for polybenzoxazines.

16.       In Table 2 the data indicated that the authors used thermal treatment programs of heating either 2 or 4 hours for each temperature? Why. Most of the previous studies were conducted the polymerization of benzoxazine monomers of only one hour for each temperature.

17.       Line 351-352: “All of the values of contact angle were about or less than 80o, indicating the moderately hydrophobic surfaces of poly(FAEN-Bz).” I disagree with this statement because it is known that hydrophobic nature is considered for contact angles greater than 90o.  

18.       Line 352-354: With increasing the curing temperatures, the values of contact angle decreased, which indicated that the hydrophobic characterization weakened and the hydrophilic characterization increased” This statement is conflicted with the polybenzoxazine nature because the previous studies reported that hydrophobicity of polybenzoxazines increases with an increase in the curing temperatures. Why the present system shows opposite behavior?

19.       Figure 11, the pictures of contact angle is not clear; therefore, they should be revised.

Author Response

Reviewer 4: Benzoxazine containing fluorinated aromatic ether nitrile linkage: Preparation, curing kinetics and dielectric properties   The manuscript describes the preparation of benzoxazine containing fluorinated aromatic ether nitrile linkage. Curing behaviors and curing kinetics of designed monomers was also studied. The obtained polybenzoxazine exhibits an excellent material and the results are very promising. The manuscript losses the scientific English writing style. Therefore, this manuscript is not yet ready to be published in this journal. There are questions/suggestions/corrections that the authors should consider before publication: 1. Lines 28 – 31: “With the development of science and technology, thermosetting resins play an important role in electronics, substrate materials, aeronautics-aerospace industry and new energy applications, due to the advantages including high specific strength and compressive strength, corrosion resistance, good mechanical and electrical properties, and outstanding bonding properties [1-4].” This sentence is very long; therefore, it should be rewritten in simple sentences. Answer to reviewer: Thanks for your comments and suggestions. According to your suggestion, this sentence was modified as “With the development of science and technology, thermosetting resins play an important role in electronics, substrate materials, aeronautics-aerospace industry and new energy applications. Thermosetting resins, like epoxy, cyanate ester, phthalonitrile, bismaleimide and so on, due to the advantages including high specific strength and compressive strength, corrosion resistance, electrical properties, light weight, high glass transition temperature (Tg) and outstanding bonding properties were intensively investigated and applied in many high-tech areas.”, and marked as red in revised manuscript. 2. Lines 55 and 56: The abbreviations should be removed. Answer to reviewer: According to you suggestion. All of the abbreviations in this paragraph were removed. 3. Line 60: “Recent years, …” should be corrected to read “in recent years, …” or “Recently,….” Answer to reviewer: Thanks for your comments. The writing “Recent years, …” was modified as “Recently,….” in revised the manuscript. 4. All references are combined with previous words; therefore a space should be added before the references. For example, Line 31 “properties[1-4].” Answer to reviewer: We are sorry for that. The space was added between references and previous words in the revised the manuscript. 5. Line 86: “our work reported previously with minor modifications” the authors should avoid using pronouns. Also, this sentence should be rewritten in corrected English grammar. Answer to reviewer: This writing was modified as “according to our previous work with minor modifications”. 6. Lines 89 and 90: “Then 3-Aminophenol was added into the solution, and amino terminated monomers (FAEN-NH2) were obtained, shown in scheme 1 (b).” This sentence should be rewritten in corrected English grammar. Answer to reviewer: This sentence was corrected as “Then 3-Aminophenol was added into the solution to prepare amino terminated monomers (FAEN-NH2). The structures and preparation processes were shown in Scheme 1 (b)”. 7. Lines 96 and 97: “Finally, the resulted mixture was dried in oven at 110 oC 96 for 4 h to remove the solvents to obtain the FAEN-Bz[10].” The authors should recheck the English language grammar. Answer to reviewer: Thanks for your suggestions. We have checked the English language grammar and modified the writing as “After that, the prepared FAEN-Bz with solution was placed into oven at 110 oC for 4 h to remove the solvents to obtain the FAEN-Bz resin”. 8. Lines 106-108: “the viscosity solution was cast on a clean glass plate in an oven to evaporate the solvent with the procedure of…….” The authors should improve the English language of the sentence. Answer to reviewer: This sentence was modified with improved English language. This sentence was revised as “Then, the FAEN-Bz solution was cast onto a clean glass plate in an oven with the sequential temperature procedure at 80 oC for 1 h, 100 oC for 1 h, 120 oC for 1 h, 140 oC for 1 h, 160 oC for 2 h and 180 oC for 2 h to remove the DMAc solvent.”. 10.       Line 115: “Differential scanning calorimetric (DSC) investigated ….” is not complete sentence. Answer to reviewer: We are sorry for that. The sentence was modified as “Curing behaviors of FAEN-Bz were studied by differential scanning calorimetric (DSC, Q100, TA Instruments, Newcastle, DE, USA) under nitrogen atmosphere with flow ratio of 50 mL/min”. 11.       Line 140: “Characteristic absorption band of -NH2 was observed at 3381 cm-1.” However, the band 3381 cm-1 was assigned for intermolecular interaction as shown in Lines 142-144 “a wide absorption band at around 3381 cm-1 was observed, which can be assigned to the intermolecular association of hydroxyl bonds, generated from the trace hydrogen and nitrogen atoms in benzoxazine rings.” The authors should clarify the both sentence for assigned groups for the band at 3381 cm-1. Answer to reviewer: Thanks for your comments. We have corrected the mistakes here and modified the Figure 2 in revised manuscript. The description was revised as “Characteristic absorption band of -NH2 was observed at 3460 and 3381 cm-1” and “a wide absorption band at around 3412 cm-1 was observed, which can be assigned to the intermolecular association of hydroxyl bonds, generated from the trace hydrogen and nitrogen atoms in benzoxazine rings.”. The modified Figure 1 (Figure 7 in revised manuscript) was copied as follows. Figure 2. FTIR spectra of (a) FAEN-NH2 and (b) FAEN-Bz. 12. Line 140, A comma should be added before the word “respectively”. Answer to reviewer: We have added the comma before the word “respectively”. 13.       Line 190: “….when setting B = 1.92 and C = 1.0008”. Because the constants B and C were given; therefore, Eq. 2 should contain these constants instead of the numbers. Answer to reviewer: We have modified the Eq.2 in revised manuscript and copied as follows. ln(β_i/(T_(α,i)^B ))=Const-C(E_α/〖RT〗_α )      (2) 14.       Line 285 and 286: “The double maximum decomposition peaks of poly(FAEN-Bz) appeared at 400 oC and 540 oC, respectively.” Why this polymer shows double maximum decomposition peaks? The authors should judge their investigations. Answer to reviewer: According the investigation on thermal stability of benzoxazine and poly(arylene ether nitrile), the two maximum decomposition peaks at 400 oC and 540 oC were corresponded the decomposition of Mannich bridge structure and aromatic ester main chains [1, 2]. The references were added in the revised manuscript and copied as follows. References: 1. Rishwana, S. Shamim, A. Mahendran, and C. T. Vijayakumar. "Studies on Structurally Different Benzoxazines: Curing Characteristics and Thermal Degradation Aspects." High Performance Polymers 27, no. 7 (2015): 802-12. 2. Tang, Hailong, Jian Yang, Jiachun Zhong, Rui Zhao, and Xiaobo Liu. "Synthesis and Dielectric Properties of Polyarylene Ether Nitriles with High Thermal Stability and High Mechanical Strength." Materials Letters 65, no. 17-18 (2011): 2758-61. 15.       Lines 288 and 289: “ …. increasing the crosslinking degree cannot improve the thermal stability obviously” I disagree with this statement because the previous investigations reported that crosslinking enhances the thermal stability of polymers, especially for polybenzoxazines. Answer to reviewer: We are sorry for that. This description was revised as “…increasing the crosslinking degree cannot intensively improve the thermal stability of poly(FAEN-Bz) in this cured condition”. 16. In Table 2 the data indicated that the authors used thermal treatment programs of heating either 2 or 4 hours for each temperature? Why. Most of the previous studies were conducted the polymerization of benzoxazine monomers of only one hour for each temperature. Answer to reviewer: In this work, the thermal treatment programs were determined via the DSC testing. As the results shown in Figure 6 in manuscript, obvious exothermic peak was observed after cured at 180 oC for 2 h and the temperature at the maximum peak was about 225 oC. This indicated that the rapid curing reaction would occur at 225 oC. However, the FAEN-Bz cured at 225 oC may lead to a large amount of defects in the cured matrix. Thus, we chose the temperature 200 oC as the curing temperature to prepare the cured matrix. After curing at 200 oC for 4 h, the exothermic peak disappeared. So, the time 4 h was chosen as the thermal treatment time when cured at 200 oC. According to our precious research, the time 2 h was chosen as the post thermal treatment time at various temperatures. 17. Line 351-352: “All of the values of contact angle were about or less than 80o, indicating the moderately hydrophobic surfaces of poly(FAEN-Bz).” I disagree with this statement because it is known that hydrophobic nature is considered for contact angles greater than 90o.   Answer to reviewer: Thanks for your useful comments. We are sorry for this writing mistake here. The contact angle less than 90o indicates the surface is hydrophilic. We have corrected the “…hydrophobic surfaces of poly(FAEN-Bz)” as “…hydrophilic surfaces of poly(FAEN-Bz)” in revised manuscript. 18.       Line 352-354: With increasing the curing temperatures, the values of contact angle decreased, which indicated that the hydrophobic characterization weakened and the hydrophilic characterization increased” This statement is conflicted with the polybenzoxazine nature because the previous studies reported that hydrophobicity of polybenzoxazines increases with an increase in the curing temperatures. Why the present system shows opposite behavior? Answer to reviewer: Thanks for your useful and constructive comments. According to your comments, we have consulted the references about the contact angle and surface energy of cured benzoxazine reported previously. The contact angle (hydrophobicity) of polybenzoxazines increased with an increase in the curing temperatures [3, 4]. The hydrophobicity of polybenzoxazine was greatly affected by the intramolecular and intermolecular hydrogen bonds. Both of the formation of intramolecular and intermolecular hydrogen bonds could contribute to the increasing of contact angle. For the two kinds of hydrogen bonds, intramolecular hydrogen bonds contributed much to increase the contact angle. Thus, we have carefully retested the contact angle of the samples with several times (> 5) for each sample. The results also exhibited the trend that with increasing curing temperature the contact angle of the samples increased. We have modified the results of the previous tests on contact angle, and make more accurate explanations about the changes. The modified Figure and Table were copied as follows. As shown in Figure 2 (Figure 13 in revised manuscript) and Table 1 (Table 4 in revised manuscript), contact angle of poly(FAEN-Bz) cured at 240oC for 2h, 260oC for 2h and 280oC for 2h was 76.9o, 81.7o and 87.4o, respectively. This can be attribited to the formation of hydrogen bonds between intramolecular and intermolecular chains. According to to the FTIR spectra shown in Figure 7 and Figure 8, characteristic absorption of intramolecular and intermolecular hydrogen bonds was observed at around 3289-3555 cm-1 and 3400 cm-1, respectively. Figure2. Contact angle images of poly(FAEN-Bz) cured at various temperatures: (a) 240oC-2h; (b) 260oC-2h and (c) 280oC-2h. Table 4. Contact angle and surface energy of poly(FAEN-Bz) Sample Liquid medium Surface tension of liquid phase media [(mN/m)] Contact angle (º) Calculation method Solid surface energy [(J/m2)] 240oC water 72.800003 76.9±0.8 Fowkes method 0.02738 260oC water 72.800003 81.7±0.3 Fowkes method 0.02383 280oC water 72.800003 87.4±2.3 Fowkes method 0.01989 240oC water 72.800003 76.9±0.8 ZHU method 0.05906 260oC water 72.800003 81.7±0.3 ZHU method 0.05646 280oC water 72.800003 87.4±2.3 ZHU method 0.05310 References: 3. Liu, Juan, Xin Lu, Zhong Xin, and Chang-lu Zhou. "Surface Properties and Hydrogen Bonds of Mono-Functional Polybenzoxazines with Different N-Substituents." Chinese Journal Of Polymer Science 34, no. 8 (2016): 919-32. 4. Goto, Masahide, Yu Miyagi, Masaki Minami, and Fumio Sanda. "Synthesis and Crosslinking Reaction of Polyacetylenes Substituted with Benzoxazine Rings: Thermally Highly Stable Benzoxazine Resins." Journal Of Polymer Science Part a-Polymer Chemistry 56, no. 16 (2018): 1884-93. 19.       Figure 11, the pictures of contact angle is not clear; therefore, they should be revised. Answer to reviewer: The Figure 11 (Figure 12 in revised manuscript) was replaced by the pictures of contact angle with improved quality. The pictures are more clearly and copied as follows: Figure 13. Contact angle images of poly(FAEN-Bz) cured at various temperatures: (a) 240 oC- 2 h; (b) 260 oC- 2 h and (c) 280 oC- 2 h

Reviewer 5 Report

This manuscript reports a benzoxazine monomer possessing fluorinated aromatic ether and nitrile groups and the corresponding thermosetting resin. After a careful evaluation, I could not recommend this article for publication in the journal of POLYMERS due to less novelty and poor experimental work. Some critical comments supporting to this decision could be

1.     Benzoxazine compounds with fluorinated groups, nitrile groups, phenyl ether linkages have been widely reported. A detail review on the prior art is missed. The targeted compound has a chemical structure possessing all of the above chemical groups. The molecular design concept on this compound is not well illustrated.

2.     For a large monomer, molecular mass and elemental analysis data should be included.  

3.     Mechanical properties measured with Instron and DMA should be included to demonstrate the “flexibility” of the prepared resins.

4.     The dielectric constants of the prepared resins are not as low compared to the reported benzoxazine based resins with low-dielectric constants.

5.     Cuing kinetics and thermal degradation studies are not much meaningful, as being not compared to other benzoxazine compounds. A system studies could be of interests to provide the relation between the chemical structures of benzoxazines and their thermal properties (curing and degradation).

6. Reference: the authors should avoid too much self-citation.

Author Response

Reviewer 5

This manuscript reports a benzoxazine monomer possessing fluorinated aromatic ether and nitrile groups and the corresponding thermosetting resin. After a careful evaluation, I could not recommend this article for publication in the journal of POLYMERS due to less novelty and poor experimental work. Some critical comments supporting to this decision could be

1. Benzoxazine compounds with fluorinated groups, nitrile groups, phenyl ether linkages have been widely reported. A detail review on the prior art is missed. The targeted compound has a chemical structure possessing all of the above chemical groups. The molecular design concept on this compound is not well illustrated.

Answer to reviewer: Thanks for your comments and suggestions on our work. The review on previous researches were added and discussed in introduction section. Benzoxazine compounds with nitrile groups were synthesized in our previous work. All of the previous work indicated that the benzoxazine-based resin containing nitrile groups and allyl possessed satisfactory properties in curing processing and structural applications. For example, allyl-functional phthalonitriles-containing benzoxazine and phthalonitrile-containing benzoxazine were successfully prepared with excellent properties like wide process temperature (~80 oC), high modulus, high Tg (> 350 oC) and high thermal stability (T5% > 450 oC in N2) [1, 2]. Chen and their co-workers reported three main-chain type benzoxazine polymers with improved toughness were synthesized via bishydroxydeoxybenzoin and three kinds of aromatic diamines, respectively. As the results also showed that the thermal stability were maintained [3]. The researchers investigated various kinds of benzoxazine resins containing fluorinated groups such as bisphenol-AF [4], 1,4-tetrafluorobenzene or 4,4-octafluorobiphenylene dioxyphenylene [5, 6] in molecular chains. The results indicated that the introduction of fluorine can serve the reducing of dielectric constants and dielectric loss. In this work, we synthesized a kind of benzoxazine containing fluorinated aromatic ether nitrile linkage. The poly(arylene ether nitriles) was a kind high performance thermoplastics, due to the aromatic ether nitrile linkage poly(arylene ether nitriles) exhibited a high Tg, outstanding tensile strength and excellent thermal stability [7]. The dielectric properties of materials could be improved by the fluorinated groups in molecular chains as the literatures reported. Thus, we combined the advantages of aromatic ether nitrile linkages and the good dielectric properties of fluorinated groups, the fluorinated aromatic ether nitrile linkages were introcuced into benzoxazine to prepare a new type of benzoxazine resin.

2. For a large monomer, molecular mass and elemental analysis data should be included.

Answer to reviewer: Thanks for your comments. The molecular mass of amino terminated fluorinated aromatic ether nitrile linkage (FAEN-NH2) was characterized by GPC. As the results showed that number-average molecular weight (Mn) was 1751 g/mol. In order to further study the structures of FAEN-NH2 and benzoxazine containing fluorinated aromatic ether nitrile (FAEN-Bz), FAEN-NH2 and FAEN-Bz were characterized by 1H-NMR and FTIR in Figure 1 and Figure 2 in revised manuscript. The results of 1H-NMR and FTIR spectroscopy confirmed the designing structures FAEN-NH2 monomers and benzoxazine containing fluorinated aromatic ether nitrile (FAEN-Bz).

3. Mechanical properties measured with Instron and DMA should be included to demonstrate the “flexibility” of the prepared resins.

Answer to reviewer: According to your comments, the DMA (Q800, TA Instruments, USA) tests were supplemented. DMA was tested by the glass fiber (GF, EW170-100, Shenyang Gaote glass fiber Co., Ltd., Shenyang, China.) reinforced FAEN-Bz composite laminates. The FAEN-Bz/GF composites prepared as follows. 8 layers of GF were impregnated with FAEN-Bz solution. Then, the prepregs dried in oven at 80 oC for 20 min and 160 oC for 15 min to remove the solvent, and the dried prepregs were placed in stainless mold under 20 MPa at 200 oC for 4 h and 240 oC for 2 h to prepare FAEN-Bz/GF composites. DMA (three-point bending) was tested from 50 oC to 260 oC (5 oC/min) under air atmosphere with the frequency of 1 Hz. The results were presented as follows. According to the glass transition theory, the temperature at the maximum peak of tan δ curve can be defined as glass transition temperature Tg. The Tg of FAEN-Bz/GF composites cured at 200 oC for 4 h and 240 oC for 2 h exhibited a high Tg (about 233 oC). Compared to the polybenzoxazine previously reported [8], polyFAEN-Bz possessed good thermal stabilities.

Figure 1. tan δ of DMA tests. FAEN-Bz/GF composites with curing procedure 200 oC/4 h and 240 oC/2 h.

4. The dielectric constants of the prepared resins are not as low compared to the reported benzoxazine based resins with low-dielectric constants.

Answer to reviewer: The initial dielectric constants of FAEN-Bz cured at 240 oC, 260 oC and 280 oC were 3.6, 4.1 and 5.0, respectively. The dielectric constants decreased with increasing the frequency. In this work, the cured FAEN-Bz resin did not exhibit low dielectric constants can be explained as follows. As the results of FTIR and DSC, hydroxyl generated from the ring-opening polymerization of oxazine rings can form the H-bonds between intermolecular and intramolecular chains which can decrease the dielectric constants. But the polar nitrile groups existed in the matrix resin were not involved in the polymerization of FAEN-Bz under the curing procedure. Thus, cured FAEN-Bz resins exhibited relative high dielectric constants.

5. Cuing kinetics and thermal degradation studies are not much meaningful, as being not compared to other benzoxazine compounds. A system studies could be of interests to provide the relation between the chemical structures of benzoxazines and their thermal properties (curing and degradation).

Answer to reviewer: Thanks for your comments. In the section 3.2.1, the gel time of FEAEN-Bz was measured at different temperature. According to the gel time, the reaction activity (Eα) was calculated via Arrhenius equation. As the results showed that, the average values (Eα) of FAEN-Bz was calculated as 98.7 kJ/mol, which was slightly lower than that of benzoxazine (102-116 kJ/mol) reported. This can be attributed to the fact that electron withdrawing groups (nitrile groups and fluorine atoms) have induced the π electron cloud of benzene slightly shifted to O and N atoms in oxazine rings, which accelerated the polymerization of FAEN-Bz, which leaded to the value of Eα was slightly lower than the benzoxazine previously reported. According to the thermal degradation studies, decomposition mechanism of cured FAEN-Bz was studied to reflect polymerization mechanisms and the intimate structures of poly(FAEN-Bz). Thus, study on cuing kinetics and thermal degradation studies were meaningful for this work.

6. Reference: the authors should avoid too much self-citation.

Answer to reviewer: Thanks for your suggestions. We have modified the references and replaced the most of our references by the works published recently.

Our academic editor's comments:

I suggest that you ask to the authors to revise and answer to comments, especially 1 and 3:

Answer to editor: Thanks for your comments and recognition for our work. The comments were carefully answered especially for No.1 and No.3. All the answers were presented above.

References:

1.           Xu, M., K. Jia, and X. Liu. "Polybenzoxazines Derived from Nitrile- and Phthalonitrile-Functional Benzoxazines and Copolymers from Benzoxazine/Phthalonitrile Resin Mixtures." In Advanced and Emerging Polybenzoxazine Science and Technology, 343-56, 2017.

2.           Xu, Mingzhen, Kun Jia, and Xiaobo Liu. "Self-Cured Phthalonitrile Resin Via Multistage Polymerization Mediated by Allyl and Benzoxazine Functional Groups." High Performance Polymers 28, no. 10 (2016): 1161-71.

3.           Chen, Chien-Han, Ching-Hsuan Lin, Jia-Min Hon, Meng-Wei Wang, and Tzong-Yuan Juang. "First Halogen and Phosphorus-Free, Flame-Retardant Benzoxazine Thermosets Derived from Main-Chain Type Bishydroxydeoxybenzoin-Based Benzoxazine Polymers." Polymer 154 (2018): 35-41.

4.           Pattharasiriwong, Patcharat, Chanchira Jubsilp, Phattarin Mora, and Sarawut Rimdusit. "Dielectric and Thermal Behaviors of Fluorine-Containing Dianhydride-Modified Polybenzoxazine: A Molecular Design Flexibility." Journal Of Applied Polymer Science 134, no. 33 (2017).

5.           Kobzar, Ya L., I. M. Tkachenko, V. N. Bliznyuk, E. V. Lobko, O. V. Shekera, and V. V. Shevchenko. "Synthesis and Characterization of Fluorinated Isomeric Polybenzoxazines from Core-Fluorinated Diamine-Based Benzoxazines." Polymer 145 (2018): 62-69.

6.           Kobzar, Yaroslav L., Ihor M. Tkachenko, Eugenia V. Lobko, Oleg V. Shekera, Anna P. Syrovets, and Valery V. Sheychenko. "Low Dielectric Material from Novel Core-Fluorinated Polybenzoxazine." Mendeleev Communications 27, no. 1 (2017): 41-43.

7.           Tang, Hailong, Jian Yang, Jiachun Zhong, Rui Zhao, and Xiaobo Liu. "Synthesis and Dielectric Properties of Polyarylene Ether Nitriles with High Thermal Stability and High Mechanical Strength." Materials Letters 65, no. 17-18 (2011): 2758-61.

8.           Kobzar, Yaroslav L., Ihor M. Tkachenko, Valery N. Bliznyuk, and Valery V. Shevchenko. "Fluorinated Polybenzoxazines as Advanced Phenolic Resins for Leading-Edge Applications." Reactive & Functional Polymers 133 (2018): 71-92.

Round 2

Reviewer 1 Report

Overall, the authors have edited a lot of errors in this revised manuscript. This paper could be accepted by following suggestions:

The reference format should follow the format of Polymers journal. 

Did the authors provide 13C NMR for benzoxazine?

Where is the storage and loss modulus of DMA analyses?

Author Response

Reviewer 2: Overall, the authors have edited a lot of errors in this revised manuscript. This paper could be accepted by following suggestions: 1. The reference format should follow the format of Polymers journal. Answers to reviewer: Thanks for your suggestions and comments. We have corrected the reference format based on the format of Polymers journal in revised manuscript. 2. Did the authors provide 13C NMR for benzoxazine? Answers to reviewer: Thanks for your useful suggestions and we quite agree with your opinion. Actually, the analysis of 13C NMR can be applied to study the structure of benzoxazine monomers more accurately. Unfortunately, in this work the testing of 13C NMR was not involved due to the limitation of test platform. Thus, the structure benzoxazine was characterized by 1H NMR and FTIR spectra. According to your constructional and positive suggestions, the 13C NMR and other characterization methods will be applied to study the structures of the monomers in our future research work. 3. Where is the storage and loss modulus of DMA analyses? Answers to reviewer: We are sorry for that. The storage and loss modulus of DMA analyses have been added in Figure S1 and copied as follows. Figure S1. Storage modulus, Loss modulus and tan δ of DMA tests. FAEN-Bz/GF composites with curing procedure 200 oC/4 h and 240 oC/2 h.

Reviewer 5 Report

GPC is not a suitable instrumnet for small molecules. The molecular weight of the prepared compounds should be measured with a Molecular Mass equipped with suitable ion sources. 

Elemental analysis is required to demonstrate the purity of the prepared compounds.

Could the TMA measurements be carried out on the neat resin, rather than its glass fiber composites?

As all the reported crosslinked benzoxazines possessing -OH groups, the answer to the point 4 shown in the previous comments is not acceptable.  

Author Response

Reviewer 5

Comments and Suggestions for Authors

1. GPC is not a suitable instrument for small molecules. The molecular weight of the prepared compounds should be measured with a Molecular Mass equipped with suitable ion sources.

Answers to reviewer: We would like to express our appreciation for your constructive advices and comments. We are sorry for the characterization of molecular weight was not measured with a Molecular Mass equipped with suitable ion sources. However, the characterization equipment was difficult to be obtained. Thus, GPC was applied to characterize the molecular weight instead of the method the reviewer mentioned. The number-average molecular weight (Mn) and weight average molecular weight (Mw) were 1751 and 2104 g/mol, with the polydispersity (Mw/Mn) of 1.20.

2. Elemental analysis is required to demonstrate the purity of the prepared compounds.

Answers to reviewer: We are sorry for that the elemental analysis was not taken into account in this work. The elemental analysis could be further applied to study the purity of the prepared compounds, and can help to improve the quality of the research work. Thus, the elemental analysis will be applied to study the purity of prepared compounds in our next work.

3. Could the TMA measurements be carried out on the neat resin, rather than its glass fiber composites?

Answers to reviewer: Thanks for your comments. It can be known that the DMA measurements were widely applied to study the relaxation of molecular chains of thermosetting matrix resin. Thus, the DMA was used to characterize the relaxation of cured benzoxazine matrix in this work. On the other hand, the preparation of pure cured benzoxazine matrix for TMA tests was difficult. So, we chose the glass fiber-reinforced benzoxazine composites to investigate the thermal mechanical properties to study the relaxation of molecular chains of cured benzoxazine. The storage modulus and loss modulus of the composites was added and shown as follows.

Figure 1. Storage modulus, Loss modulus and tan δ of DMA tests. FAEN-Bz/GF composites with curing procedure 200 oC/4 h and 240 oC/2 h.

4. As all the reported crosslinked benzoxazines possessing -OH groups, the answer to the point 4 shown in the previous comments is not acceptable.

Answers to reviewer: We are sorry for making you confusion here. As you mentioned above, -OH groups were existed in crosslinked benzoxazines via the ring-opening polymerization of oxazine rings. With the generation of -OH groups, the H-bonds could be formed between the hydrogen atoms of -OH groups and nitrogen atoms, oxygen atoms generated from the ring-opening polymerization of oxazine rings. The H-bonds were not only existed in intramolecular chains, but also existed in intermolecular chains. In this condition, the -OH groups in cured matrix were not transformed into other structures. The -OH groups and H-bonds existed in matrix were shown in Scheme 1 (Scheme 2 in revised manuscript). Finally, we are sorry for the incomplete explanation in point 4.

Scheme 2. Possible polymerization processes of FAEN-Bz: (a) the initiation of ring-opening polymerization, (b) the polymerization of oxazine rings at 180 oC, (c) the formation of H-bonds inter/intra molecular and (d) the possible structure of final polymers.

Round 3

Reviewer 5 Report

No revision has been made on this version. Molecular mass and elemnetal analyzer are common instruments for molecular charcaterization.